# Clinical Trial for Evaluating the Effectiveness and Safety of a New Dental Plaque Removal Device: Microscale Mist Unit

**DOI:** 10.3390/antibiotics11060825

**Published:** 2022-06-19

**Authors:** Hiroki Hihara, Kuniyuki Izumita, Misato Iwatsu, Tomoya Sato, Ryo Tagaino, Kenta Shobara, Yuta Shinohara, Takanori Hatakeyama, Chie Kayaba, Mariko Sato, Ayako Tokue, Tomoko Sugawara, Kanamai Ashino, Koji Ikeda, Jun Aida, Keiichi Sasaki

**Affiliations:** 1Division of Advanced Prosthetic Dentistry, Tohoku University Graduate School of Dentistry, 4-1 Seiryo-machi, Aoba-ku, Sendai 980-8575, Japan; me.cinnamoroll.whrt@gmail.com (M.I.); tomoya.sato.e3@tohoku.ac.jp (T.S.); ryo.tagaino.e1@tohoku.ac.jp (R.T.); kenta.shobara.b8@tohoku.ac.jp (K.S.); pino0121@gmail.com (Y.S.); takanori.hatakeyama.d1@tohoku.ac.jp (T.H.); keiichi.sasaki.e6@tohoku.ac.jp (K.S.); 2Perioperative Oral Health Management, Tohoku University Graduate School of Dentistry, 4-1 Seiryo-machi, Aoba-ku, Sendai 980-8575, Japan; kuniyuki.izumita.c7@tohoku.ac.jp; 3Department of Development Promotion, Clinical Research, Innovation and Education Center, Tohoku University Hospital, 1-1 Seiryo-machi, Aoba-ku, Sendai 980-8574, Japan; chie.kayaba.d2@tohoku.ac.jp (C.K.); mariko.sato.d8@tohoku.ac.jp (M.S.); ayako.tokue.e2@tohoku.ac.jp (A.T.); tomoko.sugawara.c6@tohoku.ac.jp (T.S.); kanami.ashino.d8@tohoku.ac.jp (K.A.); koji.ikeda.d6@tohoku.ac.jp (K.I.); 4Department of Oral Health Promotion, Graduate School of Medical and Dental Sciences, Tokyo Medical and Dental University, 1-5-45 Yushima, Bunkyo-ku, Tokyo 113-8510, Japan; aida.ohp@tmd.ac.jp

**Keywords:** dental biofilm, clinical trial, oral care, oral mucosa, medical device

## Abstract

This study evaluates the effectiveness and safety of a microscale mist unit (MSM-UNIT) that sprays high-speed fine water droplets to remove dental plaque adhering to the oral mucosa (tongue and palate) and tooth surface. Fifteen patients who had difficulty self-managing sufficient oral care were included in this study. Effectiveness was evaluated for at least five patients’ tongues, palate mucosas, and tooth surfaces, and safety evaluation was conducted at all three sites for all patients. Effectiveness was evaluated using the rate of degree of dental plaque removal. Safety was evaluated using a numerical rating scale (NRS) for pain and symptoms of inflammation. An operator who performed treatment and an evaluator who evaluated effectiveness and safety were designated. In addition, an image judgment committee judged effectiveness. Although evaluation of the tongue varied between the evaluators and the image judgment committee, the rates of degree for all plaque removal increased in all regions. In addition, low pain NRS results and minimal symptoms of inflammation were observed and within an acceptable range. The MSM-UNIT can be used effectively and safely for removing oral plaque not only from teeth, but also from the oral mucosa.

## 1. Introduction

Dental plaque causes not only dental caries and periodontal disease, but also aspiration pneumonia, endocarditis, and fever [1,2]. The removal of dental plaque is reported to help prevent respiratory infections in older adults, who often require a long-term caregiver [3,4,5]. It is also reported to prevent postoperative infection of hospitalized patients, contributing to reduced hospitalization periods [6,7]. Therefore, dental plaque removal is important for both oral and general health. However, older adults who require long-term nursing care and hospitalized patients often do not have sufficient oral care, and a large amount of plaque is found to adhere to the tooth surface and oral mucosa. In particular, in bedridden older persons without oral intake who are receiving nursing care, oral membranous substances composed of inflammatory cells and bacteria are frequently observed [8,9,10]. These substances attach firmly to the oral mucosa. Plaque can be removed from the tooth surface using a toothbrush or from the oral mucosa with a sponge brush. As plaque removal using these devices is associated with a risk of aspiration pneumonia, a special technique is required that is a burden for caregivers. New methods for removing oral plaque were studied and shown to be effective, including a plasma jet [11], ultrasonic activated water [12,13], and a water jet [14]. However, the application of these devices is limited to the teeth, and there are no reports investigating the effect of these methods on the oral mucosa. Recently, air abrasion devices also became available for plaque removal [15,16,17,18,19,20], and it was recognized that their proper use will not result in harmful effects on the oral mucosa [21]. However, the use of these devices is also limited to the areas around the tooth surface and gums and requires trained skills. Moreover, because these devices require extraoral suction, they are difficult to use at the bedside. No devices have regulatory approval for removing dental plaque on the whole oral mucosa in Japan.

To overcome the problems of removing plaque film in these patients, we developed a technology for plaque removal by injecting a small amount of water droplets of average diameter (≤40 µm) that are turned into mist at high pressure (few MPa) and high speed (≥100 m/s), which we termed the “Microscale Mist Unit” (MSM-UNIT; Figure 1).

Using high-speed imaging, the plaque removal mechanism of the MSM-UNIT was verified; droplets with a high kinetic state removed the artificial plaque by pushing it aside. No harmful effects occurred because of the extremely low mass of the droplets [22]. In addition, an in vitro study demonstrated that there were no harmful effects on simulated mucosa or tooth surfaces, and the technique had the same effectiveness for plaque removal as air abrasion devices [23]. Our clinical studies (registration number: UMIN000026097, date of first registration: 16 February 2017, and registration number: UMIN000031232, date of first registration: 13 February 2018) using prototype devices confirmed the safety and effectiveness of plaque removal from the palatal mucosa and tooth surface. However, the in vitro study was not a complete nonclinical model, and prior clinical studies were conducted on only a limited number of patients and conditions. To obtain regulatory approval, additional clinical studies are needed.

Therefore, the purpose of this study is to evaluate the clinical effectiveness and safety of plaque removal using the MSM-UNIT not only on teeth, but also on the oral mucosa, and to obtain data for regulatory approval.

## 2. Results

This study was conducted from March 2019 to August 2019. The submission of this research was delayed because a new patent application was under process for the MSN-UNIT following the clinical trial. Written informed consent was obtained from all 17 patients, and 15 patients were enrolled. An evaluation of effectiveness was performed for 7 patients on the tooth surface (mean age 65.7 ± 7.9 years), 6 patients on the palate (mean age 72.0 ± 6.4 years), and 12 patients on the tongue (mean age 62.3 ± 12.1 years; see Table 1 and Figure 2).

### 2.1. Effectiveness

Figure 3 and Figure 4 show examples of the application of the MSM-UNIT.

### 2.2. Plaque Removal Rate by O’Leary’s PCR

The mean PCR before treatment with the MSM-UNIT was 72.3% (48.45−96.15%), whereas after treatment it was 4.08% (-3.07–11.14%). The mean removal rate of PCR was 68.22% (48.07–88.36%; Table 2).

### 2.3. Rate of Degree of Plaque Removal (Palate and Tongue)

The removal rate of the palate plaque with an adherence degree of “1 or less” (“none” (0) or “low” (1)) after treatment was 100.0% (47.8–100.0%) at any evaluation. However, the rate for tongue plaque with an adherence degree of “1 or less” after treatment was 41.7% (15.2−72.3%). This changed to 25.0% at 1 day after treatment and 33.3% at 1 week after treatment (Table 3).

The evaluation of the degree of plaque adhesion on the palate was consistent between evaluators and the image evaluation committee. However, the committee evaluated the degree of tongue plaque adhesion to be higher than the evaluators did.

### 2.4. Plaque Removal Rate by Binarization

The removal rate by binarization was 80.47% for the tooth surface and 96.3% for the palate (Table 4).

### 2.5. Plaque Removal Time (Teeth, Palate and Tongue)

The plaque removal times for the surface of the tooth, palate, and tongue were 163.3 ± 26.8 s, 58.2 ± 30.3 s, and 65.0 ± 51.2 s, respectively (Table 5).

### 2.6. Questionnaire Survey for MSM-UNIT Treatment

Of the patients tested, when comparing the MSM-UNIT with other treatment methods, 80% reported in the survey that they were “satisfied” or “slightly satisfied.” In addition, 100% of patients provided a rating higher than “average.” Additionally, 80% of patients reported on the survey that they were “satisfied” or “slightly satisfied” with the comfort provided by MSM-UNIT-based treatment, and 93.3% patients provided a rating higher than “average”. In the survey regarding pain and irritation during treatment, 26.7% patients reported that they suffered pain and irritation, although the pain and irritation were considered tolerable (Table 6).

### 2.7. Safety Evaluation

Four patients experienced spontaneous pain in the teeth and gingiva on day 0 (after the final injection), with a mean spontaneous pain NRS score of 0.3 ± 0.5 (maximum value: 1). Two patients experienced spontaneous pain on day 1, with a mean score of 0.1 ± 0.4 (maximum value: 1). In contrast, one patient had spontaneous pain in the palate on day 0, with a mean score of 0.1 ± 0.5 (maximum value: 2). One patient also had spontaneous pain in the tongue on day 0, with a mean score 0.1 ± 0.3 (maximum value: 1). The day after treatment, two patients experienced spontaneous pain in the tooth and gums, with a mean of 0.1 ± 0.4 (Table 7).

An increase in GI score was observed in two patients on day 0 after treatment. In addition, two patients experienced an adverse event of palate bleeding on day 0 after treatment. There were no adverse events on days 1 or 7.

## 3. Discussion

In this clinical trial, the MSM-UNIT demonstrated sufficient effectiveness and safety for plaque removal, not only from the tooth surface, but also from the oral mucosa.

O’Leary’s PCR method is commonly used in Japan to evaluate dental plaque on the tooth surface, and it was considered appropriate to adopt this technique for our research because it is consistent with our clinical situation and the plaque evaluation method used in this study. Few reports are published regarding plaque removal in soft tissues, and few researchers evaluated the removal effectiveness. However, in these previous studies, the effect was assessed using only subjective methods such as macroscopic evaluation. Therefore, for objective evaluation, we performed binarization on the tooth surface and palate, and all cases were judged based on an image evaluation committee as a third-party image. Therefore, we consider the method that we used in the present study to be comparatively equivalent or more objective than those used in other studies. As O’Leary’s PCR is widely used in Japan, we investigated the effectiveness of the MSM-UNIT on the teeth in terms of plaque reduction rate. Kinoshita et al. reported that gingival health can be maintained when O’Leary’s PCR was 20% or less, which is a guideline for oral hygiene evaluation [22]. In all subjects, we observed that the PCR was less than 20% within 3 min of using the MSM-UNIT. Our experimental results matched those of our previous clinical studies and nonclinical evaluations [23] and fully demonstrated the expected performance. With regard to the conventional method of cleaning teeth using a toothbrush, Conn et al. reported that O’Leary’s PCR removal rate for a dental hygienist student was approximately 30% [24]. Furthermore, Yonenaga et al. showed that the removal rate of PCR after 3 min using a wet sheet was 10%, and 8% using a sponge brush [25]. Nobre et al. showed that an electric toothbrush can remove 50.24% of plaque in older adults [26]. Therefore, because the removal rate of PCR by the MSM-UNIT was 68.22%, the MSM-UNIT can be considered to be more effective than conventional methods.

There are many reports regarding the effectiveness of oral cleaning devices for plaque removal. However, the evaluation methods are literature specific, and there is no consensus in particular on dental plaque on the oral mucosa. There are currently no medical devices intended for plaque removal on the oral mucosa, such as the palate and tongue. Therefore, this study did not have a control group, and we used a new evaluation method incorporating visual effectiveness. Removal of plaque from the oral mucosa is important for both patients in nursing homes and hospitalized patients. Yonezawa et al. reported that using a sponge brush for oral plaque removal on the oral mucosa decreased the amount of Candida albicans [27]. Yadav et al. showed that the number of bacteria on the oral mucosa was reduced by gargling chlorhexidine [28]. Tashiro et al. also reported that bacteria in the pharynx were reduced by wiping the oral mucous with a sponge brush soaked in chlorhexidine [29]. Furthermore, Nishiyama et al. showed that professional care for 20 min once per week, including mucosa and tongue cleaning, reduces the amount of *mutans streptocci* and *Candida* species [30]. In this study, the percentage of patients who had a degree of plaque adherence of “1 or less” (“none” [0] or “low” [1]) on the palate after treatment was 100.0% at any time during the evaluation, and the removal rate of binarization was 96.3% at the palate. The judgments of both the evaluators and the image evaluation committee were almost the same. Therefore, the evaluation method was reliable and suggested that the MSM-UNIT was useful and effective for removing plaque from the palate.

In contrast, the effectiveness results for the tongue did not match the expected performance. There was a discrepancy in the ratings between the evaluators and the image evaluation committee, in which the committee rated the degree of linguistic plaque adhesion higher than the evaluators did. In many cases, in order to avoid a vomiting reflex, the operator did not spray near the base of tongue. The evaluators evaluated only the area where the mist was sprayed, whereas the image evaluation committee evaluated the whole area of the tongue. This difference in evaluation area was considered to cause the discrepancy in results, indicating that the evaluation method needed to be specified in more detail. It is known that in conventional methods that apply tools for the mechanical removal of plaque, removing the tongue coating is associated not only with reduced inflammation, but also a reduction in bad breath [31,32,33]. Some studies show that chlorhexidine is useful for reducing bacteria on the oral mucosa [27,34,35]. In addition, a previous study reported that plaque removal from the tongue is effective for those on a ventilator and reduces the burden on caregivers [36]. As the evaluation methods used for plaque removal from the tongue are also literature specific, it was judged visually in this study, similar to other studies. The evaluation method in this study was also considered to be more reliable, as it included not only an evaluator, but also the image evaluation committee as a third party. The results from the sprayed areas on the tongue suggest that the method is effective for plaque removal. In addition, Berbe et al. found that the total oral care time for nursing home residents was 37 min, in which cleaning alone took 7.4 min and oral care, including the mucosa and tongue, took 20 min [37]. As a result of the MSM-UNIT requiring a shorter time to remove plaque, it is considered not only to be effective for patients at home, but also for reducing the burden on nursing home operators. Additionally, the MSM-UNIT is operated with a handpiece and a foot pedal and can be used without stress because dental workers are familiar with using this type of equipment. Therefore, the training time required to use the MSM-UNIT is considered the same as that required for conventional techniques, such as chlorhexidine swabs. In particular, our questionnaire survey showed that patients were more satisfied with the MSM-UNIT than they were with conventional methods. Although the cost of introducing and maintaining the MSM-UNIT is higher than that of swabs, the usage time (as mentioned above) is relatively low. This is a sufficient benefit for not only operators, but also for patients. Additionally, the primary feature of the MSM-UNIT is the removal of plaque using only a small amount of water. Thus, costs can be reduced because other chemicals or toothpastes are not re-quired. Therefore, given the cost/benefits of the MSM-UNIT, we consider it to be of equal or superior value to conventional techniques.

Although five adverse events were observed, these were grade 1 events with no treatment required. The most frequent adverse event was gingivitis (13.3%). The area of gingivitis was limited, and for this reason it was evaluated as “unrelated” to the MSM-UNIT. Occurrences of gingival pain (6.7%), tongue bleeding (6.7%), and oral bleeding (6.7%) were evaluated as “related” to the MSM-UNIT; however, these adverse events were transient and resulted in no clinical problems. The questionnaire survey revealed that pain and irritation were tolerable and NRS was 2 at maximum; thus, there were no clinical problems with pain.

In addition, the results were considered to be within a sufficiently acceptable range compared with the adverse events of chlorhexidine, which includes allergies, soreness, irritation, mild desquamation, and mucosal ulceration/erosions [38]. Therefore, the risk of using the device was considered to be less than that associated with conventional oral care. As there were no symptoms the next day or one week later, it is considered that treatment with the MSM-UNIT can be safely performed on the palate and tongue once a week.

Oil pulling and swabs are traditionally used for oral hygiene. No harmful effects or improvement in oral health were reported with these methods [39]. Recently, pro- and para-probiotics were tested for use in for oral hygiene, and studies report the effectiveness and safety of these oral care methods [40,41]. Although the most important type of oral healthcare is self-care, this is difficult for patients with serious medical conditions, such as those who are bedridden or have dementia. Therefore, a method of plaque removal performed by a dental worker is also needed, and the demand for this type of technique is expected to increase as the population of older people increases. The primary feature of the MSM-UNIT is the removal of plaque using only a small amount of water, thereby reducing costs due to the absence of other chemicals or toothpastes. Moreover, the method is also simpler and safer than other conventional techniques. Additionally, the MSM-UNIT can remove oral plaque easily and safely, and it may prevent aspiration pneumonia in elderly people and those in nursing homes and hospitals. The handpiece of the MSM-UNIT can be sterilized, and its other parts can be disinfected with alcohol; thus, the possibility of cross-infection with other patients is extremely low. However, the handpiece must be replaced for each patient, which is costly. Additionally, the device requires a small amount of water, and its use causes aerosol release; thus, its application might be limited because of the need for suction. To evaluate the preventive effect of aspiration pneumonia, another study should be conducted to observe the long-term prognosis using the MSM-UNIT.

## 4. Materials and Methods

Before starting this study, we obtained protocol consultations from the Pharmaceutical and Medical Device Agency (PMDA), a regulatory authority in Japan, and clinical trial notifications. In those consultations, it was observed that the use of the device is simple, usage methods across medical institutions have almost no influence on the results of effectiveness and safety, and it was considered not to affect the evaluation. Additionally, a split-mouse design is difficult because features of the mist can be removed and the effect might be exerted beyond the set area. Self-care is not appropriate in a control group setting because the device is used by an operator, and there is a high possibility that the subject will be biased because of changes in motivation due to study participation. In addition, in Japan, no devices have clinical approval for removing oral plaque on the whole oral mucosa. This clinical trial was conducted to obtain clinical approval for the MSM-UNIT in Japan. Therefore, from the perspective of regulatory science, using devices that are not approved in Japan in the comparison group was judged to be inappropriate due to uncertainty in term of effectiveness and safety. Additionally, the procedures used in conventional methods that use a toothbrush, tongue scraper, or sponge brush are difficult to standardize because of operator dependence. As such, it is difficult to set a control group, there is a high possibility that the control group will be biased, and before-and-after treatment comparisons within the same subject are preferable for evaluating the device’s effectiveness and safety. Therefore, this study was designed as a single-arm, open-label, single-center, within-subject clinical trial in accordance with consultation with the PMDA.

The protocol of this study was approved by the Tohoku University Hospital Institutional Review Board (reference No. 183004) in accordance with the Declaration of Helsinki. This study was consistent with the Good Clinical Practice (GCP) guideline. In addition, this study was registered on the University Hospital Medical Information Network (UMIN). The registration number was UMIN000035950 (date of first registration: 01 March 2019, https://center6.umin.ac.jp/cgi-bin/ctr/ctr_view_reg.cgi?recptno=R000040945, accessed on 21 March 2022). Inclusion criteria were as follows: (1) patient age greater than 20 years; (2) patient signed a written consent form; (3) the case was evaluated for effectiveness on the tooth surface, with the presence of at least 5 remaining teeth in one jaw and at least 10 remaining teeth in both jaws, except for the third molar; and (4) the patient was judged to have significant plaque on the tongue, palatal mucosa, tooth surface, or at least two of these sites.

The exclusion criteria of patients were as follows: (1) the case was evaluated for efficacy on the tooth surface and for the presence of malocclusion; (2) extensive tooth restoration was required during the study period; (3) presence of at least 6 mm probing pocket depth and bleeding on probing periodontitis in more than half of the teeth; (4) presence of grade 3 Miller classification of tooth mobility; (5) presence of moderate hypersensitivity; (6) presence of malignant tumor, leukoplakia, lichen planus, oral candida, tongue pain, or other diseases judged by the dentist as affecting the evaluation; (7) presence of acute inflammation; (8) the patient was unable to stop taking analgesic drugs 24 h before the start of the evaluation; (9) participation in another clinical trial; (10) presence of a severe vomiting reflex; and (11) history of aspiration pneumonia.

Fifteen patients were included in this study. The effectiveness was evaluated in at least five patients’ tongues, palate mucosas, and tooth surfaces; two or more sites were evaluated in some patients. The safety evaluation was conducted at all three sites for all patients. The primary objective of this trial was to obtain regulatory approval. Therefore, in consultation with the PMDA, we set the patients with palate-attached plaque as the worst case from a regulatory science perspective. Effectiveness and safety could be evaluated by macroscopic evaluation if at least five patients were included. Through a similar approach, effectiveness and safety could be evaluated if at least five patients were included in other parts. Therefore, the minimum sample size needed to obtain regulatory approval was fifteen patients.

### Treatment Protocol

#### Application of the MSM-UNIT

Spray conditions were set with a water flow rate of 10 mL/min and an air pressure of 0.2 MPa. The tip of the handpiece was positioned at least 6 mm above the target and moved for 60 s so that it did not spray a single site for more than 2 s.

To eliminate bias as much as possible during the evaluation period in this clinical trial, we designated an operator who performed treatment using the MSM-UNIT and an evaluator who evaluated effectiveness and safety. The operator and evaluator were different people and did not change their role during the clinical trial period. Although the operator sprayed all parts of the tongue, palate, and tooth surface, only areas rated for a plaque adhesion degree of more than “high” were included for evaluation of effectiveness. Domains that were not evaluated for effectiveness were treated for safety evaluation. The degree of plaque adhesion was judged as “very high” (3), “high” (2), “low” (1), and “none” (0), with reference to the judgment sample (Figure 5).

Figure 6 presents a chart of the treatment flow. The treatment was conducted in sequence of the tongue, palate, and tooth surface. The treatment was stopped at the point at which the degree of plaque adhesion was rated as “none” (0) or <20% of O’Leary’s plaque control record (PCR) [42,43] within 60 s. The treatment was performed up to three times until the degree of adhesion was rated as “none” (0) or <20% of the PCR (the total treatment time was as long as 180 s). Safety evaluation domains were treated for 60 s. Palate and teeth were stained before treatment using Plaque Check Gel BR (GC). The safety assessment was conducted 1 day after (day 1) and 1 week after (day 7) the treatment.

## 5. Analysis

### 5.1. Analysis of Effectiveness Evaluation

The evaluators judged the scores by comparing sample photographs. In addition, a member of the image judgment committee evaluated whether the evaluators’ judgment was appropriate. Members of the committee had no conflicts of interest with this trial.

Multiple primary endpoints were set in the study. As a single endpoint did not provide an overall treatment effect, this trial was characterized by assessing treatment effects over multiple dimensions.

### 5.2. Plaque Removal Rate by O’Leary’s PCR (Teeth)

We calculated the summary statistics of the PCR and the rate of changes before and after removal (number of instances, mean, 95% confidence interval for mean, standard deviation).

### 5.3. Rate for Degree of Plaque Removal (Palate and Tongue)

We calculated the percentage of plaque adherence degree for “low” (1) or less (“low” (1) or “none” (0)) and 95% confidence interval of the percentage.

### 5.4. Plaque Removal Rate by Binarization

We analyzed images of the teeth and palate that had “high” (2) or higher plaque adhesion at the time of eligibility confirmation. The validity of the cleaning rate was preanalyzed by the image judgment committee using Adobe Photoshop CC. Fixed teeth were excluded from the analysis because they are difficult to binarize. The region of interest (ROI) was set for natural teeth, which can be confirmed for front, left-side, and right-side views. The ROI area (pixel value), plaque adherence area (pixel value), and cleanup rate were set after calibration. Summary statistics of the removal rate and calibrated removal rate for the plaque adherence area were calculated (number of instances, mean, 95% confidence interval for mean, and standard deviation).

### 5.5. Plaque Removal Time (Teeth, Palate and Tongue)

Summary statistics were computed for the time taken to remove all plaque from the tongue and palate and PCRs less than 20% on tooth surface.

## 6. Subject Satisfaction with MSM-UNIT Treatment

To compare the MSM-UNIT with other treatment methods, we administered a questionnaire survey to the subjects regarding their satisfaction with both treatment and comfort, which they rated according to five grades (“dissatisfied”, “slightly dissatisfied”, “average”, “slightly satisfied”, and “satisfied”). Additionally, a survey regarding pain and irritation during the treatment was administered. Subjects recorded the presence or absence of pain, and in the event of pain, the site and degree were rated according to four grades (“tolerable”, “tolerable within 3 min”, “tolerable within 1 min”, and” intolerable”).

### Safety Assessment

The summary statistics of spontaneous pain of the teeth, gums, palate and tongue according to the numerical rating scale (NRS) were calculated. A frequency aggregation of changes in the gingival index (GI) and inflammation symptoms was also performed. Adverse events were also recorded.

## 7. Conclusions

The results of this study show that the MSM-UNIT device can be used to effectively and safely remove plaque from the tooth surface and oral mucosa (palate). With regard to the tongue, despite the discrepancy between the judgments of evaluators and the image judgment committee, the device was considered to be effective because of the overall reduction in plaque. Therefore, the MSM-UNIT can be used in whole oral care to effectively and safely remove plaque.

## Figures and Tables

**Figure 1 antibiotics-11-00825-f001:**
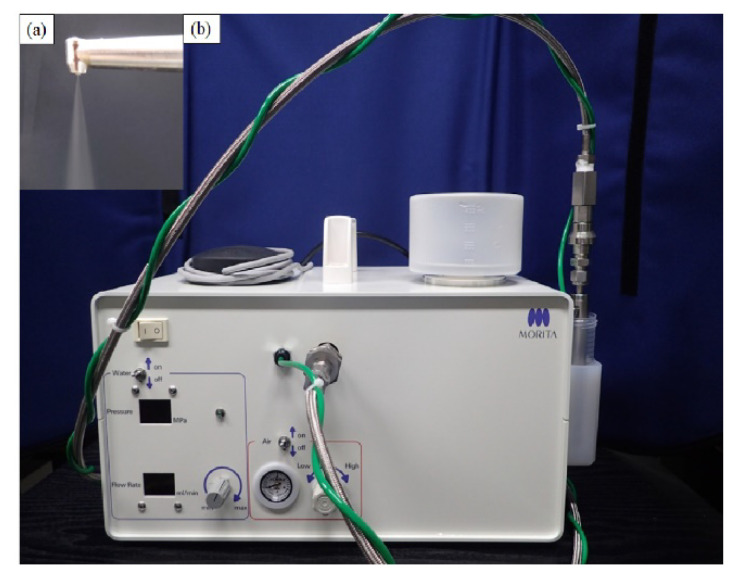
Photographic images of the MSM-UNIT spraying with (**a**) a handpiece and (**b**) the main body.

**Figure 2 antibiotics-11-00825-f002:**
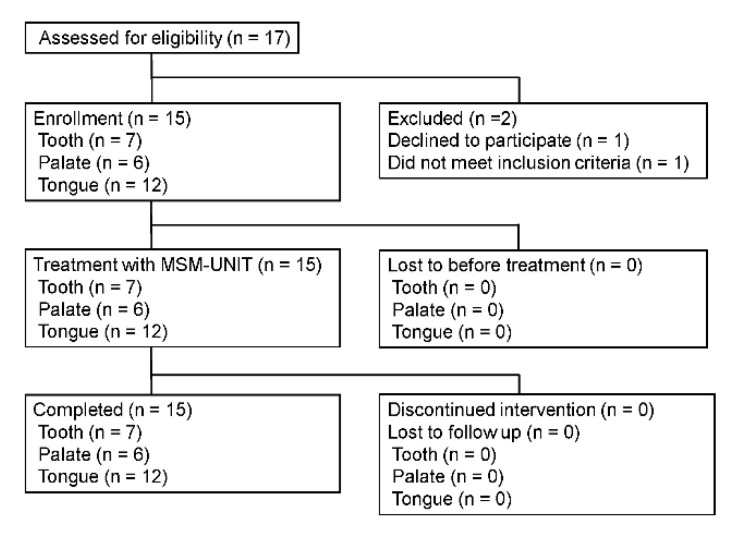
Flow chart of the study’s subjects.

**Figure 3 antibiotics-11-00825-f003:**
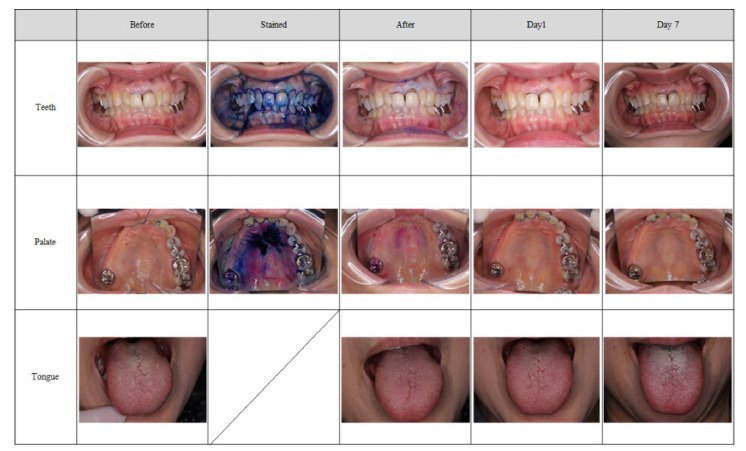
Teeth, palate, and tongue application results of the MSM-UNIT. Before treatment, stained, after treatment, day 1 and day 7 are shown.

**Figure 4 antibiotics-11-00825-f004:**
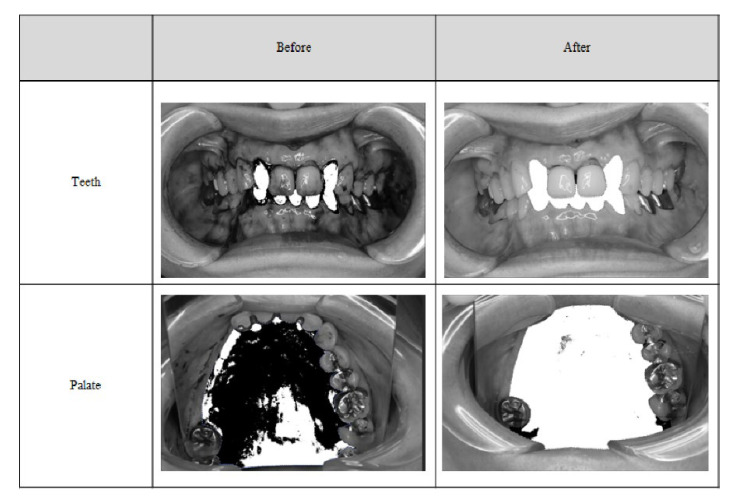
Teeth and palate photographs before and after the binarization.

**Figure 5 antibiotics-11-00825-f005:**
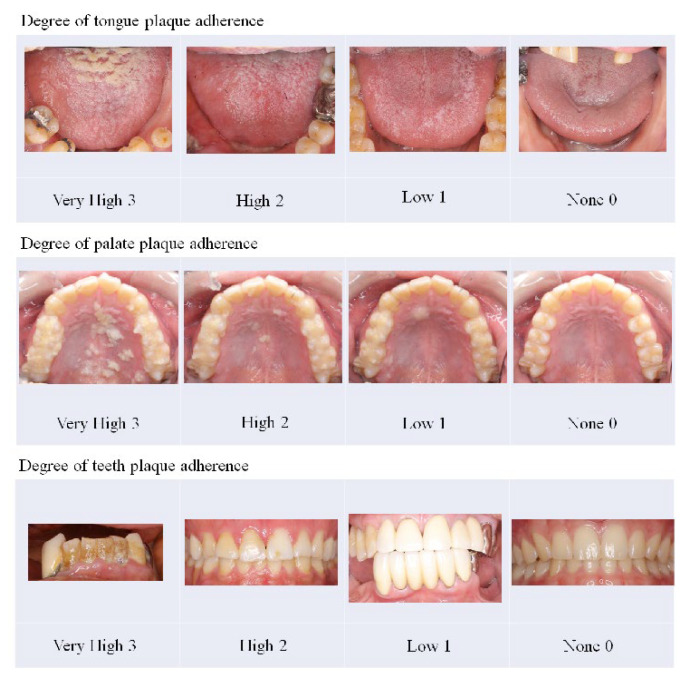
Judgment sample of the degree of plaque adhesion. Each area was evaluated as one of four grades: “very high” (3), “high” (2), “low” (1), and “none” (0).

**Figure 6 antibiotics-11-00825-f006:**
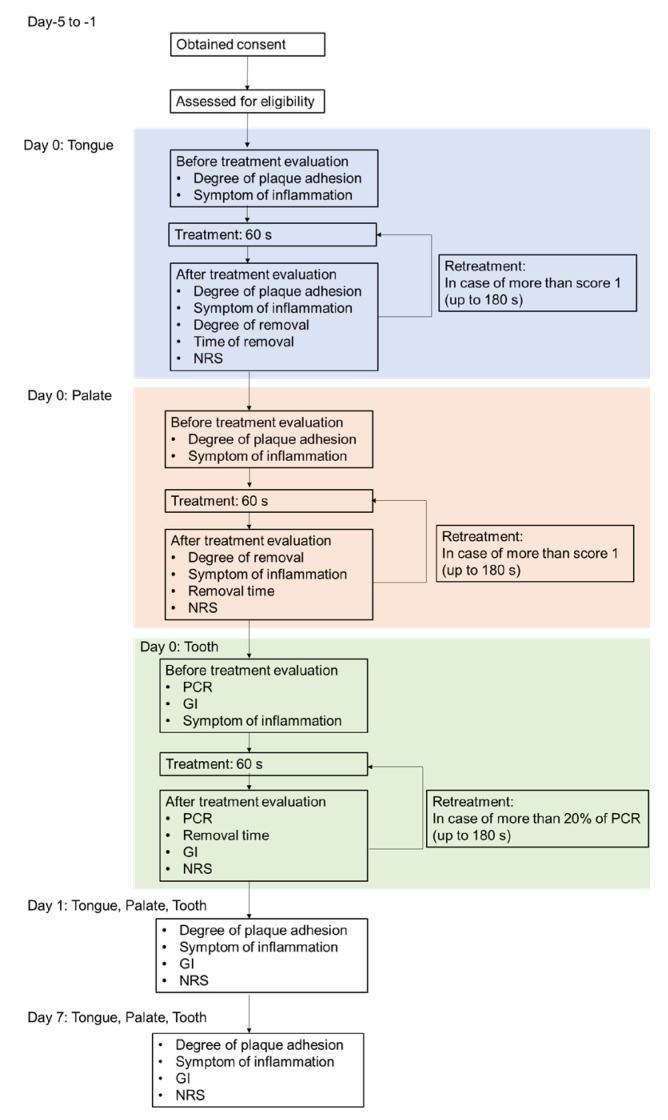
Chart of the treatment flow.

**Table 1 antibiotics-11-00825-t001:** Patient data.

	Teeth	Palate	Tongue
Number of patients	6	5	12
Male (*n*)	3	0	6
Female (*n*)	3	5	6
Mean age (years)	65.7	72.0	62.3

**Table 2 antibiotics-11-00825-t002:** Plaque removal rate by PCR (% Plaque removal rate by PCR (%)).

	Before Treatment	After Treatment	Amount of Change
*n*	6	6	6
Mean (95% confidence interval)	72.30 (48.45–96.15)	4.08 (−3.07–11.24)	68.22 (48.07–88.36)
Standard deviation	22.72	6.82	19.20

**Table 3 antibiotics-11-00825-t003:** Evaluation of the degree of palate and tongue plaque adherence.

		None 0	Low 1	High 2	Very High 3	Low or None	95% Confidence Interval
Palate	Before treatment	0	0	4	1	0.0% (0/5)	0.0–52.2%
	After treatment	3	2	0	0	100.0% (5/5)	47.8–100.0%
Tongue	Before treatment	0	0	6	6	0.0% (0/12)	0.0–26.5%
	After treatment	0	5	7	0	41.7% (5/12)	15.2–72.3%

**Table 4 antibiotics-11-00825-t004:** Plaque removal rate by binarization (teeth and palate).

	Teeth	Palate
*n*	6	5
Mean (95% confidence interval)	80.47 (68.28–92.65)	96.38 (92.23–100.53)
Standard deviation	11.61	3.34

**Table 5 antibiotics-11-00825-t005:** Plaque removal time (teeth, palate, and tongue).

	Teeth	Palate	Tongue
*n*	6	5	12
Mean ± standard deviation (seconds)	163.3 ± 26.8	58.2 ± 30.3	65.0 ± 51.2

**Table 6 antibiotics-11-00825-t006:** Questionnaire survey results for MSM-UNIT treatment.

Satisifaction compared with other treatment methods	Satisfied		53.3% (8)		
	Slightly		26.7% (4)		
	Average		20.0% (3)		
	Slightly dissatisfied		0.0% (0)		
	Dissatisfied		0.0% (0)		
	Not clear		0.0% (0)		
	Other		0.0% (0)		
Comfortabiltity	Satisfied		40.0% (6)		
	Slightly		40.0% (6)		
	Average		13.3% (2)		
	Slightly dissatisfied		6.7% (1)		
	Dissatisfied		0.0% (0)		
	Not clear		0.0% (0)		
	Other		0.0% (0)		
Pain and irritation	There was no pain and irritation		73.3% (11)		
	There was pain and irritation		26.7% (4)		
			Teeth	Palate	Tongue
		Tolerable	0	1	0
		Tolerable within 3 min	0	0	1
		Tolerable within 1 min	1	1	0
		Intolerable	0	0	0
	Not clear		0.0% (0)		
	Other		0.0% (0)		

**Table 7 antibiotics-11-00825-t007:** Spontaneous pain as rated on the NRS.

		Teeth/Gum	Palate	Tongue
Before treatment	Mean ± standard deviation (*n*)	0.0 ± 0.0 (15)	0.0 ± 0.0 (15)	0.0 ± 0.0 (15)
	min, median, max	0, 0.0, 0	0, 0.0, 0	0, 0.0, 0
After treatment	Mean ± standard deviation (*n*)	0.3 ± 0.5 (15)	0.1 ± 0.5 (15)	0.0 ± 0.0 (15)
	min, median, max	0, 0.0, 1	0, 0.0, 2	0, 0.0, 0
Day 1	Mean ± standard deviation (*n*)	0.1 ± 0.4 (15)	0.0 ± 0.0 (15)	0.1 ± 0.3 (15)
	min, median, max	0, 0.0, 1	0, 0.0, 0	0, 0.0, 1
Day 7	Mean ± standard deviation (*n*)	0.0 ± 0.0 (15)	0.0 ± 0.0 (15)	0.0 ± 0.0 (15)
	min, median, max	0, 0.0, 0	0, 0.0, 0	0, 0.0, 0

## Data Availability

Restrictions apply to the availability of these data. Data were obtained from J. MORITA MFG. CORP and AMED, and are available from the authors upon reasonable request and with the permission of J. MORITA MFG. CORP and AMED.

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
