# Peer review of "Clinical Trial for Evaluating the Effectiveness and Safety of a New Dental Plaque Removal Device: Microscale Mist Unit"

_antibiotics, 2022, doi:10.3390/antibiotics11060825_

Round 1

Reviewer 1 Report

This is a well written paper concerning the efficacy of a new device to reduce oral plaque both on hard & soft tissues. It is unusual to have the results & discussion appear prior to methodology. The study was conducted in mid 2019 and an explanation would be appropriate as to reasons for manuscript delay. 

With regards to the methods:

- the explanation given for the lack of a self-control or not using a  split mouth design would aid methodological justification

- the failure to use one or more comparitor plaque removal devices needs justification e.g using an electric plaque removal device with changeable heads for tongue & other soft tissues other than teeth 

- power calculations were not included & this needs justification

- the very small sample size together with the reliance on subjective measures of plaque accumulation, in addition to the large standard deviation [table 2] weakens the strength of results

- a patient evaluation would be beneficial & strengthen results

With regards to discussion:

- This is rather brief and needs to more fully explain the validation of the plaque evaluation strategies used and how this relates to best evidence & other studies.

- only chlorhexidine was discussed for plaque removal but what about essential oil and coconut oil pulling & swabs?

- the discussion fails to discuss this device's user friendliness and how a non-dentally trained user could use it safely & without creating aspirations

- the discussion could include a more convincing argument as to why this device is better than improving the effectiveness of patients current plaque removal regime

- there should be included a cost/benefit analysis 

- a devices ease/difficulty in terms of infection control could be discussed

With regards to results:

- tables 3 & 4 could be combined into a single table 

Author Response

Thank you very much for your insightful letter. We appreciate the reviewers for their valuable suggestions, especially those on how to improve our content. We have attempted to address the comments raised by the reviewers as follows:

Comment 1:

This is a well written paper concerning the efficacy of a new device to reduce oral plaque both on hard & soft tissues. It is unusual to have the results & discussion appear prior to methodology. The study was conducted in mid 2019 and an explanation would be appropriate as to reasons for manuscript delay

Response to comment:

Thank you for your question. The results and discussion precede the methodology because the journal guidelines have stipulated the methodology to be the penultimate section of the main text. The reason for the submission delay is that a new patent application occurred for this device after the clinical trial. We have provided that reason in the Results section.

Comment 2

- the explanation given for the lack of a self-control or not using a split mouth design would aid methodological justification

Response to comment:

Thank you for your suggestion. As mentioned in the manuscript, the protocol was designed as per the guidelines of the Pharmaceutical and Medical Device Agency (PMDA), the Japanese regulatory authority for obtaining clinical notification. As for the control group setting, self-care was not appropriate because this device is used by an operator, and there is a high possibility that the subject will be biased due to changes in motivation by virtue of their participation in the study. Additionally, since the feature of the mist can be removed and the effect may be exerted beyond the set area, the split-mouse design was difficult to implement. We have provided this reason in the Materials and Methods section.

Comment 3

the failure to use one or more comparitor plaque removal devices needs justification e.g using an electric plaque removal device with changeable heads for tongue & other soft tissues other than teeth

Response to comment:

Thank you for your suggestion. There are no clinically approved devices for removing an oral plaque from the entire oral mucosa. This clinical trial was conducted to obtain the clinical approval of MSM-UNIT in Japan. Therefore, from the regulatory science perspective, setting devices that have not been approved in Japan as controls was judged to be inappropriate because it was compared with devices whose efficacy and safety were uncertain. We have mentioned that reason in the Materials and Methods section.

Comment 4

power calculations were not included & this needs justification

Response to comment:

Thank you for your comment. Unfortunately, no power analysis was performed to calculate the sample size. Since the main goal of this trial was to obtain regulatory approval. Therefore, in consultation with the PMDA, we set the patients with plaques attached to the palate as the worst case from the viewpoint of regulatory science, and the effectiveness and safety can be evaluated by macroscopic evaluation if at least five patients are included. With the same viewpoint, effectiveness and safety can be evaluated if at least five patients were included in other parts. Therefore, a minimum sample size of 15 patients was needed to obtain regulatory approval. We have provided that reason in the Materials and Methods section.

Comment 4

the very small sample size together with the reliance on subjective measures of plaque accumulation, in addition to the large standard deviation [table 2] weakens the strength of results

Response to comment:

Thank you for your suggestions. As you have indicated, the sample size seems small; however, we think it shows that plaque removal was possible. We would like to make it a future issue, including the calculation of the sample size.

Comment 5

a patient evaluation would be beneficial & strengthen results

Response to comment:

Thank you for your suggestion. In this study, we performed within-patient before-treatment and after-treatment comparisons. As mentioned above, the standard deviation increased as the study was performed with the minimum possible number of patients.

Comment 6

With regards to discussion:

- This is rather brief and needs to more fully explain the validation of the plaque evaluation strategies used and how this relates to best evidence & other studies.

Response to comment:

Thank you for your suggestion.

Regarding the plaque evaluation method in this study, in Japan, O’Leary’s PCR is a common technique for the evaluation of dental plaques on tooth surfaces, and it is appropriate to adopt it because it is consistent with the clinical situation. There are few reports on plaque removal in soft tissues and few have evaluated the removal effect; however, in those studies, it was only assessed using subjective methods, such as macroscopic evaluations. Therefore, for objective evaluations, in this study, binarization is performed on the tooth surface and palate, and all cases are judged by the image evaluation committee as a third-party image. Therefore, it is either comparable to or more objective than other studies. We have mentioned that reason in the Discussion section.

Comment 7

only chlorhexidine was discussed for plaque removal but what about essential oil and coconut oil pulling & swabs?

Response to comment:

Thank you for your suggestion. As you have indicated, essential oil and coconut oil pulling and swabs improve oral health. We have described the comparison with essential oil and coconut oil pulling and swabs in the Discussion section and have added a reference.

Comment 8

the discussion fails to discuss this device's user friendliness and how a non-dentally trained user could use it safely & without creating aspirations

Response to comment:

Thank you for your suggestion. MSM-UNIT is operated using a handpiece and a foot pedal. It can be used without stress because dental workers are familiar with its use. Conversely, as you have indicated, non-dental workers need training; however, it is assumed that MSM-UNIT will be used by dental workers only. We have provided that reason in the Discussion section.

Comment 9

the discussion could include a more convincing argument as to why this device is better than improving the effectiveness of patients current plaque removal regime

Response to comment:

Thank you for your suggestion.

Although self–oral care is most important for oral health, it is difficult for patients, such as bedridden patients and those with dementia. Therefore, a method of plaque removal by dental workers is also needed and the demand is expected to increase with the aging society. The greatest feature of MSM-UNIT is plaque removal with a small amount of water. The cost can be reduced in the absence of any other chemicals or toothpaste that can be used more easily and safely than other conventional methods. We have provided this reason in the Discussion section.

Comment 10

there should be included a cost/benefit analysis

Response to comment:

Thank you for your suggestion. As mentioned in our response to comment #9, using MSM-UNIT reduced the cost. We made this argument in the Discussion section.

Comment 11

a devices ease/difficulty in terms of infection control could be discussed

Response to comment:

Thank you for your insightful suggestion. MSM-UNIT can remove oral plaques easily and safely, and it may prevent aspiration pneumonia in elderly people and patients in nursing homes and hospitals. However, this device requires a little water, and its use could be limited by the need for suction. To evaluate the preventive effect of aspiration pneumonia, another study aimed at observing the long-term prognosis using MSM-UNIT is required. We made this argument in the Discussion section.

Comment 12

With regards to results:

- tables 3 & 4 could be combined into a single table

Response to comment:

Thank you for your insightful suggestion. We have combined these tables and changed the numbers of other tables.

Reviewer 2 Report

Dear Authors, 

thank you for your contribution dealing with the effectiveness and safety of a new dental plaque removal device.

I think that your work is well conducted and relevant from a scientific point of view. Moreover the photographic documentation is really interesting.

The only issue in my opinion is related to the bibliography. I think that you should delete old references and add some more recent.

Moreover, I think that a fundamental aspect that deserves to be discussed is the use of probiotics/paraprotics for the domicialiary maintenance of oral hygiene. In fact, this aspect constitutes a hot topic in dental hygiene research. You should refer to the reseach by Butera, Scribante et al.

After that I think that the manuscript can be considered for publication in the journal-.

Yours sincerely,

the Reviewer 

Author Response

Thank you very much for your insightful letter. We appreciate the reviewers for their valuable suggestions, especially those on how to improve our content. We have attempted to address the comments raised by the reviewers as follows:

Comment 1

Thank you for your contribution dealing with the effectiveness and safety of a new dental plaque removal device.

I think that your work is well conducted and relevant from a scientific point of view. Moreover the photographic documentation is really interesting.

Response to comment:

Thank you for your compliments.

Comment 2

The only issue in my opinion is related to the bibliography. I think that you should delete old references and add some more recent.

Response to comment:

Thank you for your insightful suggestion. Particularly, the references about the air polishing device were old; thus, we replaced these reports with new ones.

Comment 3

Moreover, I think that a fundamental aspect that deserves to be discussed is the use of probiotics/paraprotics for the domicialiary maintenance of oral hygiene. In fact, this aspect constitutes a hot topic in dental hygiene research. You should refer to the reseach by Butera, Scribante et al.

Response to comment:

Thank you for your suggestion. As you have indicated, probiotics and paraprotics are also important for oral health. We have described the comparison with probiotics and paraprotics and have also added the appropriate references.

Round 2

Reviewer 1 Report

I thank the authors for addressing most of the suggestions following review of the paper. There does appear to be areas where my suggestions were possibly unclear or misinterpreted and were therefore not adequately addressed.

With reference to the comment regarding a patient evaluation, this was referring to whether any evaluation had been done in terms of the patient's perspective as to the effectiveness of this product. In other words, did the patient consider the product effective and comfortable to have been used in their mouths, how painful was it from their perspective, were there any issues such as mouth burning, inflammation or other sensory effects reported by patients? If there was no patient perspective evaluation done, this needs to be justified as well as listed as a major limitation.

With reference to the cost benefit analysis, this comment referred to a much broader consideration of cost versus benefit such as the cost to purchase such a unit (for example, if the cost of the unit is thousands of dollars it may not be affordable), how much time/cost would be required to train someone in using the unit, any other particular costs that would be specific to this device (e.g. any disposables or consumables) and justifying the cost of purchasing & using such a device compared to costs/benefits over current practice such as chlorhexidine swabs which are very inexpensive & require little if any training.

With reference to the infection control issues, I was referring more to infection control surrounding the unit itself, given it would be used in hospital wards and on immuno-vulnerable patients, how the unit would be disinfected, how any tubing and handpieces would be disinfected, sterilised or are they disposable, as well as consideration of issues with the aerosol creation and surface contamination with possible cross-infection with other patients. 

It would be appreciated if these matters were better considered within the manuscript.

Author Response

Thank you for your insightful letter. We appreciate the reviewers for their valuable suggestions, especially those on how to improve the content of our manuscript. Please find below our point-by-point responses to the comments raised by the reviewers.

Reviewer1

Comment 1:

I thank the authors for addressing most of the suggestions following review of the paper. There does appear to be areas where my suggestions were possibly unclear or misinterpreted and were therefore not adequately addressed.

With reference to the comment regarding a patient evaluation, this was referring to whether any evaluation had been done in terms of the patient's perspective as to the effectiveness of this product. In other words, did the patient consider the product effective and comfortable to have been used in their mouths, how painful was it from their perspective, were there any issues such as mouth burning, inflammation or other sensory effects reported by patients? If there was no patient perspective evaluation done, this needs to be justified as well as listed as a major limitation.

Response to comment:

Thank you for your explanation and questions, and I apologize for the misunderstanding.

We evaluated the perspective of the patients using a patient satisfaction survey. This has been described in the “Subject satisfaction with MSM-UNIT treatment” sections (Methods section 4.2.5. and Results section 2.1.5.). A questionnaire survey was conducted on patient comfort and a comparison with other treatment methods. We revised the Results section because the original description was difficult to understand and provided a new table in this section. In the survey, 80% of the subjects reported that they were “satisfied” or “slightly satisfied” in terms of Comfortability and Satisfaction in comparison to other treatment methods. Thus, from the patients’ perspective, the MSM-UNIT is considered effective due to its high satisfaction rating compared with those of other treatments.

We also evaluated the perspective of the patients in terms of pain and irritation. However, we determined that NRS scores were more objective; hence, we did not include these results. Unfortunately, we did not ask the patients about the quality of the pain, but we assessed how much pain they could tolerate. Pain and irritation caused by the MSM-UNIT were tolerable to the patients: the maximum NRS score was 2, which is considered clinically acceptable. We have provided an appropriate explanation in the Discussion section.

Comment 2

With reference to the cost benefit analysis, this comment referred to a much broader consideration of cost versus benefit such as the cost to purchase such a unit (for example, if the cost of the unit is thousands of dollars it may not be affordable), how much time/cost would be required to train someone in using the unit, any other particular costs that would be specific to this device (e.g. any disposables or consumables) and justifying the cost of purchasing & using such a device compared to costs/benefits over current practice such as chlorhexidine swabs which are very inexpensive & require little if any training.

Response to comment:

Thank you for your explanation. As you point out, the cost of introducing and maintaining the MSM-UNIT is higher than that of swabs. However, the training time required to use the MSM-UNIT is the same as that required for conventional techniques, such as chlorhexidine swabs, and the usage time is reduced considerably relative to that of conventional techniques. Additionally, our questionnaire survey showed that patients were more satisfied with the MSM-UNIT than they were with conventional methods. Thus, the benefits are sufficient for not only the operators but also the patients. Furthermore, the principal feature of the MSM-UNIT is the removal of plaque using only a small amount of water, which reduces costs by removing the need for other chemicals and/or toothpaste. Therefore, based on this cost/benefit analysis, we considered the use of the MSM-UNIT to be equally valid or more beneficial relative to conventional techniques. We have provided an appropriate explanation in the Discussion section regarding the same.

Comment 3

With reference to the infection control issues, I was referring more to infection control surrounding the unit itself, given it would be used in hospital wards and on immuno-vulnerable patients, how the unit would be disinfected, how any tubing and handpieces would be disinfected, sterilised or are they disposable, as well as consideration of issues with the aerosol creation and surface contamination with possible cross-infection with other patients.

Response to comment:

Thank you for your explanation. The handpiece of the MSM-UNIT can be sterilized, and the other parts can be disinfected with alcohol. The device requires a small amount of water, and its use causes aerosol release, which can be removed using intraoral suction. The handpiece is replaced for each patient; thus, the risk of cross-infection is extremely low. We have provided an appropriate explanation in the Discussion section.

Reviewer 2 Report

Dear Authors,

thank you for your improvements to the manuscript.

I think that all my requests have been addressed.

There is an only think which must be corrected and it is reference number 42. The reference as reported is uncortect.

The correct reference is as follows:

Butera A, Gallo S, Maiorani C, Molino D, Chiesa A, Preda C, Esposito F, Scribante A. Probiotic Alternative to Chlorhexidine in Periodontal Therapy: Evaluation of Clinical and Microbiological Parameters. Microorganisms. 2020 Dec 29;9(1):69. doi: 10.3390/microorganisms9010069. PMID: 33383903; PMCID: PMC7824624.

Yours faithfully 

The reviewer

Author Response

Thank you for your insightful letter. We appreciate the reviewers for their valuable suggestions, especially those on how to improve the content of our manuscript. Please find below our point-by-point responses to the comments raised by the reviewers.

Comment 1:

I think that all my requests have been addressed.

There is an only think which must be corrected and it is reference number 42. The reference as reported is uncortect.

The correct reference is as follows:

Butera A, Gallo S, Maiorani C, Molino D, Chiesa A, Preda C, Esposito F, Scribante A. Probiotic Alternative to Chlorhexidine in Periodontal Therapy: Evaluation of Clinical and Microbiological Parameters. Microorganisms. 2020 Dec 29;9(1):69. doi: 10.3390/microorganisms9010069. PMID: 33383903; PMCID: PMC7824624.

Response to comment:

Thank you for pointing out this error. We have revised the reference accordingly

Round 3

Reviewer 1 Report

The authors are to be commended for addressing reviewers suggestions thereby improving the manuscript.